# Impact of Movement Control Training Using a Laser Device on the Neck Pain and Movement of Patients with Cervicogenic Headache: A Pilot Study

**DOI:** 10.3390/healthcare11101439

**Published:** 2023-05-15

**Authors:** Songui Bae, Juhyeon Jung, Dongchul Moon

**Affiliations:** 1Department of Physical Therapy, Graduate School, Dong-Eui University, Busan 47340, Republic of Korea; 2Department of Physical Therapy, College of Nursing, Healthcare Sciences and Human Ecology, Dong-Eui University, Busan 47340, Republic of Korea; 3Department of Physical Therapy, Gimhae College, Gimhae-si 50811, Republic of Korea

**Keywords:** cervicogenic headache, neck pain, movement control training

## Abstract

This study verified the effect of movement control training using a laser device on the neck pain and movement of patients with cervicogenic headache. A total of twenty outpatients recruited from two Busan hospitals were equally divided into two groups. The experimental group underwent movement control training with visual biofeedback, while the control group performed self-stretching. Both groups received therapeutic massage and upper cervical spine mobilization. A four-week intervention program was also conducted. Measurement tools including the cervical flexion–rotation test, visual analog scale, Headache Impact Test-6, pressure pain threshold, range of motion, sensory discrimination, and Neck Disability Index helped assess the participating patients before and after the intervention. Additionally, the Wilcoxon signed-rank test and the Mann–Whitney U test helped determine inter and intra-group variations, respectively, before and after the intervention. Most of the measurement regions revealed significant changes post-intervention within the experimental group, while only the cervical flexion–rotation test, visual analog scale, Headache Impact Test-6, and Neck Disability Index indicated significant changes post-intervention within the control group. There were also considerable inter-group differences. Thus, movement control training using a laser device more effectively improves neck pain and movement of patients with cervicogenic headache.

## 1. Introduction

Headaches affect approximately 66% of the global population, and these can lead to disability and reduced work productivity [1]. Cervicogenic headache (CGH) is a type of secondary headache caused by a cervical disorder [2]. CGH is characterized by unilateral headache involving the neck. In cases like these, a neck palpation indicates limited cervical range of motion (CROM) and upper cervical pain [3]. CGH generally occurs at the cervical 2–3 zygapophysial joint [4] and dysfunctions may be observed at the atlanto- occipital, atlantoaxial, and zygapophysial joints in the cervical spine [5,6].

Cervicogenic headache (CGH) arises from pain stimulation of the spinal nerves C1, C2, and C3 in the cervical region, which transmit pain signals to the trigeminocervical complex (TCC) [3]. Pain stimulation in the upper cervical region overlaps structurally and terminally with the pain-sensitive nerve endings of the first and second cervical regions of the trigemino-cervical complex (TCC), and this convergence can cause pain associated with the head and around the eyes [7]. Previous studies claimed that a sensory disorder arises from reduced tactile sensitivity and increased mechanical sensitivity of the trigeminal nerve as a result of such negative stimulations [8,9,10,11,12].

Previous research has reported that FHP has a significant impact on the musculoskeletal system [13]. FHP can lead to various conditions, including cervical nerve root compression, CGH, and dizziness [14]. FHP involves an increase in extension of the upper cervical spine and flexion of the lower cervical and upper thoracic spine [15]. As a result, FHP can cause instability in the upper cervical and upper thoracic regions and hypermobility in the lower cervical region, leading to muscle tension and weakness patterns known as Upper Crossed Syndrome [16,17]. This FHP can cause biomechanical symptoms in the upper cervical region and is considered the root cause of CGH due to joint instability [14].

As a result of such postures, the abnormal contractions around the head and neck induce muscle tension, causing a CGH [18]. Notably, muscles including the sternocleidomastoid (SCM), upper trapezius (UT), levator scapulae (LS), scalene, suboccipalis (SO), pectoralis minor, and pectoralis major are shortened [16,19]. Self-stretching is often used as an intervention to relax the reduced muscles by increasing the range of muscle extension to restore muscle activity and promote flexibility to improve overall performance [20]. However, although stretching is used to transiently relieve pain, its benefits are limited in cases of the long-term treatment for CGH based on corrected posture improvement [21,22,23].

Due to the association between CGH and cervical dysfunctions, the joint mobilization of the upper cervical spine is reported as the most effective intervention to improve CGH symptoms [24,25,26]. So far, various studies on CGH have shown that the joint mobilization of the upper cervical spine could lower the pain related to headaches but induce limited changes in FHP or cervical mobility. Thus, this method is effective solely in pain relief but inadequate as a treatment to improve muscle imbalance or correct postural abnormality [27].

A previous study suggested movement control training as a new method of functional improvement in patients with chronic neck pain [28]. Movement control training applied to the cervical spine is one of the cervical spine stabilization exercises that focuses on correcting repetitive and habitual wrong movements rather than muscle strengthening. This MCT maintains cervical curvature in the upper cervical spine and prevents the upper cervical spine from destabilizing or becoming FHP [28,29]. The method improved the functional component to a greater degree, as it could effectively enhance the physical, psychological, and behavioral aspects [29,30].

The reported feedback methods to promote an effective performance of the movement control training include self-palpation, tactile feedback, visual feedback, and verbal instructions from a therapist [28]. Visual feedback-based training allows an independent and accurate performance of exercise, as it provides real-time visual data [31]. Such training is considered an effective method to enhance the ability of bilateral postural control and motivate patients [32]. A rehabilitation training protocol with visual feedback allows integrated training by enhancing eye and head movements, balance exercises, visual pursuits, gaze stability, and positional control as well as reducing neck pain and postural sway [33,34,35].

Most previous studies on patients with CGH investigated the joint mobilization of the cervical spine and neck muscle stretching [22,24,25,26]. However, there is a general lack of studies that verify the effect of movement control training to correct the habitual pattern of movement in CGH patients or the related mechanical sensitivity or sensory ability.

This study hypothesizes that using a laser device for movement control training is more effective than stretching for improving pain and neck function in patients with CGH. Therefore, this study aims to compare the effects of laser device-based movement control training after a 4-week intervention in patients with CGH.

## 2. Materials and Methods

### 2.1. Study Design

The study participants included 20 willing outpatients recruited from 2 hospitals in Busan. This study used a single-blind, two-group, pre-test-post-test design and was approved by the Institutional Review Board at Dongeui University (IRB No. DIRB-202204-HR-E-08).

The participants were divided into the intervention group that would undergo movement control training and the control group that would undergo self-stretching. The intervention program was conducted for four weeks, with three sessions per week [36,37]. One session was conducted in a day, which lasted for an hour. The intervention group received therapeutic massage of the neck muscle and manual mobilization of the upper cervical spine combined with visual biofeedback-based movement control training. The control group underwent therapeutic massage of the neck muscle and manual mobilization of the upper cervical spine combined with self-stretching. To gauge the effect of the intervention, the pain, flexion-rotation test (FRT), cervical range of motion (CROM) sensory discrimination, and Neck Disability Index (NDI) were measured before the intervention (pre-test) and four weeks after the intervention (post-test). For each evaluation, the average value of the results of repeated measurements 2 to 3 times was used. Figure 1 presents the flow chart of the study design and procedures.

One experienced physical therapist with more than five years of clinical experience took measurements. The intraclass correlation coefficient (ICC 3,1) value calculates the intra-tester reliability by repeated measurements. The PPT test showed that SO was 0.985 (95% CI: 0.971 to 0.990), LS was 0.983 (95% CI: 0.971 to 0.990), and UT was 0.983 (95% CI: 0.972 to 0.990). The sensory discrimination test showed 0.816 (95% CI: 0.186 to 0.990). The FRT test showed that the right side was 0.889 (95% CI: −0.635 to 0.176), the left side was 0.867 (95% CI: −0.619 to 0.200). The ROM test showed that flexion was 0.645 (95% CI: 0.102 to 0.859), the extension was 0.604 (95% CI: −0.001 to 0.843), side bending was 0.775 (95% CI: 0.431 to 0.911), rotation was 0.853 (95% CI: 0.629 to 0.942).

### 2.2. Participants

The participants in this study were 20 individuals who voluntarily agreed to participate after receiving a detailed explanation of the study protocol. In addition, informed consent was obtained from all subjects involved in the study. All participants satisfied the inclusion criteria and were divided into the intervention (movement control training) and control groups via random sampling (Figure 1). Sixteen cards, marked “A” for the intervention group, and another 16 cards, marked ‘B’ for the control group, were placed in an opaque envelope and drawn by the participants. This investigator was not involved in the interventions and assigned the participants to random groups.

Among the patients diagnosed with CGH in reference to the diagnostic criteria of the International Headache Congress, the eligible patients satisfied the following specific criteria [38,39,40]: (1) Positive result in FRT (pain during FRT and ROM ≤ 33°), (2) Headache Impact Test-6 (HIT-6) ≥ 50, (3) Unilateral headache with the symptoms starting from the upper neck and the back of the head to the front of the eyeball on the affected side, (4) Neck ROM below the normal range (flexion: <60°, extension: <75°, lateral flexion: <45°, rotation <78°), (5) Pain caused by the movement of the neck or continuation of a single posture, (6) Pain caused by external pressure on at least a single segment across the upper cervical joints (C0-3), (7) Subjects who scored less than 40 on the central sensitization inventory test, (8) Among the subjects, those who showed faulty movement in the movement control test, of the cervical spine in three directions (flexion: overhead arm lift test, extension: horizontal retraction test, rotation: head turn test) [28].

The exclusion criteria were as follows: (1) Headache with the symptoms starting from an area other than the neck, (2) An autonomic nerve dysfunction, (3) Brain tumor, fracture, metabolic syndrome, rheumatoid arthritis, osteoporosis, or hypertension, traumatic injury of the cervical spine; (4) Headache caused by a central nervous system disorder.

For 10 participants (5 participants each in the intervention and control groups), the preliminary sample size estimated by the G*power program (ver. 3.1.9.2, Heinrich-Heine-University, Düsseldorf, Germany) under the conditions of effect size being 1.3, significance level of 0.05, and testing power of 0.8 was 18. Based on this, 20 individuals were recruited in consideration of a 10% drop-out rate.

### 2.3. Outcome Measures

#### 2.3.1. Flexion-Rotation Test

To measure the quantified level of dysfunction of the upper cervical joint in patients with CGH, the flexion-rotation test (FRT) was performed. With the participant in a supine position, the rater moved the cervical joint of the participant to the maximum flexion. The upper cervical joint was rotated to the left and to the right, with the head of the participant leaning against the abdomen of the rater. The movement was stopped when the participant showed a symptom or the rater sensed a hard edge. The CROM was measured in triplicate for each rotation using a spinal assessment device (Performance Attainment Associates, MN, USA), and the mean value was obtained. The level of pain during FRT was measured in triplicate based on the visual analog scale (VAS), and the mean value was obtained [39,41,42].

#### 2.3.2. Cervical Range of Motion

CROM was measured using a spinal assessment device (Performance Attainment Associates, Lindstrom, MN, USA). The CROM measurements were taken while flexion, extension, left and right side bending, and left and right rotation. The participant was guided to sit at 90° between the hip and knee joints, with the cervical spine in a neutral position. Subsequently, they actively performed the cervical flexion, extension, left and right side bending, and left and right rotation to the maximum range. All measurements before and after the intervention were taken in triplicate by a single rater as the participant repeated each motion thrice, and the mean value was obtained [43].

#### 2.3.3. Pressure Pain Threshold

To measure the pressure pain threshold (PPT) at the SO, LS, and UT muscles, an algometer (Baseline, Sammons Preston, Bolingbrook, IL, USA) was used. The upper border of the trapezius muscle halfway between the midline and the lateral border of the acromion, the levator scapulae muscle 2 cm above the lower insertion located in the upper medial border of the scapulae and the suboccipital points 2 cm lateral to the spinous processes of the axis were considered. The participant was guided to maintain a comfortable posture sitting on a chair, and the algometer was applied in the vertical direction at a rate of 1 lb/s. At the point at which the participant indicated the onset of pain using a verbal signal, the algometer level was measured in the unit of lb/cm^2^. The mean of triplicate measurements taken in one-minute intervals was used as the final score [44].

#### 2.3.4. Sensory Discrimination

For the sensory discrimination test, four block grids were marked on the participant’s neck, and the rater was given an image of four block grids. The goal of the rater was to identify the site stimulated by a discriminator (Jamar, Pattersen Medical, Missisauga, ON, Canada). The size of each grid in the 4-block grids was set at 42.2 mm, below the maximum distance in the 2-point discrimination on the normal 7th cervical vertebrae (50 mm) [45]; the total length of the four block grids was set to be within the distance of 1 cm below the occipital bone to the 7th cervical spinous process. The same procedure was repeated twice.

#### 2.3.5. Headache Impact Test-6

HIT-6 stands for Headache Impact Test-6, which is a tool for assessing the functional impairment of headaches using 6 questions that measure pain, social functioning, role functioning, cognitive functioning, psychological distress, and vitality. The scores of HIT-6 are distributed between the lowest 36-point to the highest 78-point, depending on the intensity of the headache. Higher scores indicate a higher intensity of headache. A score above or equal to 60 indicates a severe disorder, a score of 56–59 indicates considerable disability in daily life, a score of 50–55 indicates notable disability, and a score below or equal to 49 indicates little or no disability [40].

#### 2.3.6. Neck Disability Index

To measure the impairment of neck movement, the NDI was used as a tool consisting of 10 items from the intensity of pain to the performance of daily activities, reading, working, lifting, driving, sleep, headache, concentration, and performance of leisure activities. Each item is rated on a scale of 0–5; the total score was obtained. A score of 0–4 indicates no disability, of 5–14 indicates little disability, of 15–24 indicates moderate disability, of 25–34 indicates severe disability, and a score above or equal to 35 indicates complete disability [46].

### 2.4. Interventions

#### 2.4.1. Therapeutic Massage and Manual Mobilization

Each participant received therapeutic massage on 3 muscles (SO, SCM, and UT) for approximately 7 min, and the total duration of the massage on the neck area was 20 min. Additionally, 5-min resting period was provided after each muscle was treated, and both the right and left sides were massaged equally [47].

Manual mobilization was administered in 2 segments (C0-C1 and C2-C3) for approximately 10 min to each participant for 20 min [39,48]. An adequate amount of resting time was granted after treating each segment, and a 5-min rest was provided. Manual mobilization was administered similarly to each joint on both the left and right sides (Table 1).

#### 2.4.2. Movement Control Training Using a Laser Device

For the visual biofeedback-based movement control training that the intervention group underwent, a laser feedback device (SenMoCOR™ laser headlamp, Orthopedic Physical Therapy Products, Minneapolis, MN, USA) was positioned at the center of the forehead of the participant in a standing posture, with a straight back and arms on the side of the body. The target was set at 60 Inches from the participant’s eye level throughout the training [49].

To apply the movement control training of cervical flexion, the participant was asked not to move the head while gazing toward the front and below and maintaining the head and lower cervical spine in normal curvature; they had to lift both arms completely to a 180° shoulder flexion and then bring the arms back down to the sides of the body [28,29,34].

To conduct the movement control training of cervical extension, the participant was guided to maintain the scapular and TM joint in a neutral position while taking care not to push the chin forward (upper cervical extension) or move the head forward (lower cervical flexion); both arms had to be pulled backward at 15–20°, with the head and cervical spine in a neutral position [28,29,34].

To apply the movement control training of rotation, the participant was guided to lean against the wall and position the lower and upper cervical in normal curvature for the training. The laser was kept on a horizontal line as the participant was guided to turn the head to gaze beyond one shoulder and then make a complete contralateral turn of the head with the movement at 70–80° [28,29,34] (Table 1) (Figure 2).

#### 2.4.3. Self-Stretching

The control group participants were guided to stretch the UT and LS muscles on each side for one minute per session, this was repeated five times. Adequate resting time was provided between each session [50] (Table 1).

### 2.5. Statistical Analysis

The data collected in this study were analyzed using SPSS 22.0 (IBM Corp, New York, NY, USA) for Windows, and the level of significance for statistical validation was set at 0.05. The general characteristics of the participants were analyzed through descriptive statistics, and an independent *t*-test was performed to test the inter-group homogeneity. The Shapiro–Wilk test was used to check normal distribution. As all variables were shown to satisfy normality, a parametric method of analysis was used.

To analyze the pre-test and post-test changes between the groups, an independent *t*-test was performed. To examine the intra-group variation, the paired *t*-test was conducted.

## 3. Results

### 3.1. Characteristics of Participants

A total of twenty patients with CGH participated in this study. Table 1 shows the participants’ general characteristics and the result of the homogeneity test (Table 2).

### 3.2. Flexion-Rotation Test

The FRT result post intervention demonstrated significant changes in the ROM and VAS for the intervention group (*p* < 0.05) as well as the control group (*p* < 0.05). For the inter-group comparison, the variation between pre-test and post-test was compared, and the ROM (left side; *p* = 0.03, right side; *p* = 0.00) and VAS (left side; *p* = 0.00, right side; *p* = 0.01) displayed significant differences (Table 3) (Figure 3).

### 3.3. Cervical Range of Motion

The ROM post intervention varied significantly for the intervention group (*p* < 0.05) across all four variables. However, for the control group, only the ROM of the flexion and left and right rotation showed significant changes. For the inter-group comparison, the variation between pre-test and post-test was compared. Only the ROM of the left rotation showed a significant difference (*p* = 0.00) (Table 3) (Figure 3).

### 3.4. Pressure Pain Threshold

The PPT after the intervention varied significantly for the intervention group (*p* < 0.05) across all the tested muscles (*p* < 0.05). However, for the control group, only the right SO showed a significant change (*p* < 0.05). For inter-group comparison, the variation between pre-test and post-test was compared, and all muscles displayed a significant difference (So left side; *p* = 0.01, So right side; *p* = 0.00, LS left side; *p* = 0.00, LS right side; *p* = 0.01, UT left side; *p* = 0.00, UT right side; *p* = 0.00) (Table 4) (Figure 3).

### 3.5. Sensory Discrimination

Post intervention, a significant change in sensory discrimination was found in the intervention group (*p* < 0.05) but not in the control group (*p* > 0.05). For inter-group comparison, the variation between the pre-test and post-test was compared, and both groups exhibited a significant difference in sensory discrimination (*p* = 0.00) (Table 4) (Figure 3).

### 3.6. Headache Impact Test-6

Post intervention, the HIT-6 varied significantly in the intervention group (*p* < 0.05) as well as the control group (*p* < 0.05). For inter-group comparison, the variation between pre-test and post-test was compared, and the two groups exhibited a significant difference in the HIT-6 (*p* = 0.00) (Table 5) (Figure 3).

### 3.7. Neck Disability Index

Post intervention, the NDI varied significantly in the intervention group (*p* < 0.05) as well as the control group (*p* < 0.05). For inter-group comparison, the variation between pre-test and post-test was compared, and the two groups exhibited a significant difference in the NDI (*p* = 0.01) (Table 5) (Figure 3).

## 4. Discussion

This study aimed to verify the effect of movement control training using a laser device on neck pain and movement of CGH patients through comparisons with conventional physical therapy after four weeks of an intervention. In conclusion, CGH patients who underwent movement control training using a laser device showed improvements in FRT, ROM, PPT, sensory discrimination, HIT-6, and NDI. This study aims to verify the effect of movement control training using a laser device on neck pain and movement in CGH patients through comparison with conventional physical therapy after four weeks of intervention. This study hypothesized and conducted experiments that movement control training would improve pain, cervical spine movement, tissue mechanosensitivity, sensory discrimination, and neck function in CGH patients.

In previous studies, the quantification of upper cervical dysfunction in CGH patients had been viewed as the key method of assessment [51], and the FRT has been the method of such quantification [41]. In this study, likewise, the pain during FRT and change in the ROM significantly varied between the pre-test and post-test in the two groups. This coincided with previous studies on patients with CGH and chronic neck pain with respect to the significant improvement of pain and the ROM upon FRT when joint mobilization was conducted on C0-C1 and C2-C3 [39,52]. The reason for the improvements in the FRT result in the two groups is conjectured to be due to the intervention program consisting of upper cervical mobilization for both groups. As such, upper cervical mobilization could increase the ROM upon FRT as a method to enhance the mobility at C1-C2, which is responsible for 50% of cervical rotation [53]. Meanwhile, inter-group comparison according to pre- and post-test variation confirmed that the level of improvement was higher for the intervention group than the control group. This may be accounted for based on the increased functional variation in a previous study where a treatment combining manual therapy and exercise therapy was applied [52]. Patients with chronic neck pain underwent movement control training in a previous study, and improvements in pain and ROM were reported [30]. The movement control training applied in the study was based on the principle of the control of functional impairment and pain by securing dynamic stability via the regulation of uncontrolled movements and preventing micro-damage of surrounding tissues [28]. The movement control training applied in the intervention group involved isolated movements toward independent movements of the neck bone based on the spine as an adjacent joint, while the stress and joint pressure on surrounding tissues were reduced to promote the stability of the neck bone and normal movements. This led to significant differences in the ROM and pain between the intervention and control groups.

In clinical practice, the measurement of neck ROM is an assessment to determine the state of a patient’s joint and the effect of the therapeutic intervention, which offers diagnostic criteria for disability related to daily life [54]. The results in this study revealed that the intra-group variation in the ROM was significant across all variables for the intervention group that underwent movement control training, while only the flexion and left and right rotation showed a significant change for the control group, which underwent self-stretching. In previous studies on FHP patients based on stretching, significant changes were shown solely by the left rotation and the left and right side bending, and it was reported that the stretching led to a significant improvement in the ROM after at least four weeks and that long-term use of stretching would lead to more effective results in terms of the flexibility of soft tissues [55,56]. The increase in the ROM for certain control group participants in this study is presumed to be due to the short (four-week) intervention that limited the changes in the viscoelasticity of soft tissues. Based on these findings, stretching for four weeks or more should be included as part of an intervention to improve the ROM in all directions.

Visual biofeedback with task-oriented training was reported to improve neck ROM in a previous study [49]. The reason for the significant change in the intervention group is presumed to be due to the task-oriented training that used visual biofeedback with the cervical curvature in a neutral position and in the directions of flexion, extension, and rotation. Task-oriented training was suggested as an effective intervention in previous studies for the learning of movements and improvement of ROM through repeated exercise and practice, which lends support to this study’s findings [57,58,59].

The latest research trend shows increased interest in the change of sensory ability due to increased pain [60].

The sensory discrimination test in this study revealed that the intervention group had a significant increase of 18% observed in the intra-group variation, whereas the control group had no significant variation. In a previous study on patients with chronic neck pain, the use of biofeedback training was shown to improve the neck sensory system [61]. The improvement of sensory discrimination in the intervention group is presumed to be due to the effect of movement control training to enhance the joint position sense and the control of muscle contractions at physical segments. Tactile functions were reported in previous studies to include the ability to induce selective muscle contractions to suitable levels, the ability to reposition the joint, and the ability to control movements [20,28]. Thus, the movement control training In this study is likely to have improved the joint position sense at the neck to cause selective muscle contractions, which ultimately enhanced the tactile functions.

The level of pain was measured via the PPT and HIT-6 in this study. For the PPT, the mechanical sensitivity to pressure was analyzed for the SO, LS, and UT muscles. The increased level of pressure in the PPT was taken to indicate a fall in the mechanical sensitivity of the muscles [62]. An increase in pain was reported to affect physical movements by inducing changes in the somatosensory cortex, leading to tactile alteration [9]. The intra-group variation analyzed in this study revealed a significant increase in the pre-test and post-test variation for the intervention group across all the tested muscles.

Meanwhile, the HIT-6 was measured to estimate the intensity of headaches in daily life. The HIT-6 consists of six domains to reflect the effect of headaches on the quality of life, social functions, cognitive functions, and psychological pain; higher scores indicate higher headache intensity [40]. The intra-group variation analyzed in this study revealed a significant decrease in the post-test HIT-6 by 22% in the intervention group and by 7% in the control group. Nevertheless, the variation between the two groups was significant to suggest that movement control training using a laser device is an effective intervention to reduce the HIT-6. Hence, the movement control training using a laser device could improve the PPT and HIT-6, presumably because the perception of the change of muscle length and the sensory ability in the muscles surrounding the neck were enhanced through movement control training to reduce the pain and mechanical sensitivity.

Regarding the PPT, the control group displayed a significant increase only in the right SO. The reason stretching was applied to the control group in this study is that when joints are connected continuously and have a common direction of movement, one joint is more flexible than others and more prone to excessive movement [63]. If movement control does not occur appropriately in one joint, stiffness increases relative to adjacent joints, which limits movement [28]. In order to maintain normal function, other joints in the motor control system need to compensate for this limitation, and relatively flexible movement occurs [64]. However, if muscles are connected to relatively flexible movements, it can cause a limited range of motion in adjacent areas and uncontrolled movements [28]. Therefore, stretching was applied to the control group to make stiff tissues more flexible, prevent excessive movement in joints, and facilitate functional exercise. Sling-based stretching by FHP patients for four weeks in a previous study did not lead to a significant change in the PPT, which agreed with the result obtained for the control group in this study [65]. The reason for the significant increase only in the right SO could be due to the SO being small in size and attached by a single joint, so that the stretching applied in an accurate way could induce changes in the muscle length as well as the mechanical sensitivity. In contrast, the LS and UT are relatively larger compared to the SO and attached by multiple joints to prevent accurate stretching with a consequent lack of change in the muscle length, leading to no apparent improvement in the PPT.

The inter-group comparison in this study showed significant variation between both groups to suggest that movement control training using a laser device effectively reduces the mechanical sensitivity of surrounding tissues.

The NDI is a reliable questionnaire to assess the level of disability in patients with CGH [58]. It consists of 10 items related to the intensity of pain, daily activities, headache, concentration, work, and leisure [46]. The mean post-test NDI score was significantly lower by 6.3 in the intervention group and 3.7 in the control group. Nevertheless, the two groups varied significantly in the pre- and post-test variation to suggest that movement control training is an effective intervention.

Neck pain has a 70% correlation with Temporomandibular disorders (TMD) [66], and studies have shown that the presence of TMD increases the NDI score [67]. Several studies have investigated the relationship between cervical pain and TMD, and the presence of trigger points in the cervical muscles is very common [68,69,70]. Furthermore, MTrps in the neck and head muscles can cause headaches [71] and the presence of MTrps in the trapezius muscle can lead to TMJ imbalance and overloading of the masticatory muscles [72,73]. The masticatory muscles are innervated by the trigeminal nerve and converge with the trigeminocervical nucleus, the nociceptive nucleus of the upper cervical spine, which can be a cause of cervical headaches [74]. In the future, the treatment of TMJ should be considered in the management of cervical headaches in clinical practice.

Meanwhile, a close association between CROM and NDI was reported in a previous study [75]. The effective reduction of the NDI in the intervention group may be associated with the effect on the ROM, as it is conjectured to be due to the increased ROM in all directions.

In a previous study where patients with chronic neck pain underwent proprioceptive training and stability training, the NDI was shown to decrease significantly [76]. The movement control training in this study thus involved repetitions of movements commonly used in daily life toward enhanced proprioceptive functions and stability. The consequent change in the behavior and perception leading to pain is likely to have improved the NDI in an integrated way.

This study has some limitations. First, the sample size was small (n = 20), and the level of continuous intervention effects after four weeks could not be determined. Second, despite the close association of headaches with depression and stress, psychological factors were not incorporated into the intervention used in this study. Third, as the participants were outpatients, only the intervention time and no other environmental factors could be controlled. Fourthly, we could not control the participants’ vision, which may have affected their ability to correct their posture. Fifthly, both groups received basic interventions, such as massage and joint mobilization, and we could not compare the effects of MCT and stretching independently. Therefore, the basic intervention may have a complex effect. Hence, future studies comparing single and complex interventions are needed to address the limitations of this study and to provide more comprehensive evidence for the effectiveness of these interventions.

## 5. Conclusions

The results of this study show that movement control training using visual feedback can improve pain and neck function in patients with CGH. in clinical practice, an effective intervention to improve neck pain and function in CGH patients could entail treatment combining manual therapy and movement control training to correct habitual movement patterns. Furthermore, an intervention combining task-oriented training and movement control training would increase the movement of physical segments and enhance the sensory ability, improving neck movement.

## Figures and Tables

**Figure 1 healthcare-11-01439-f001:**
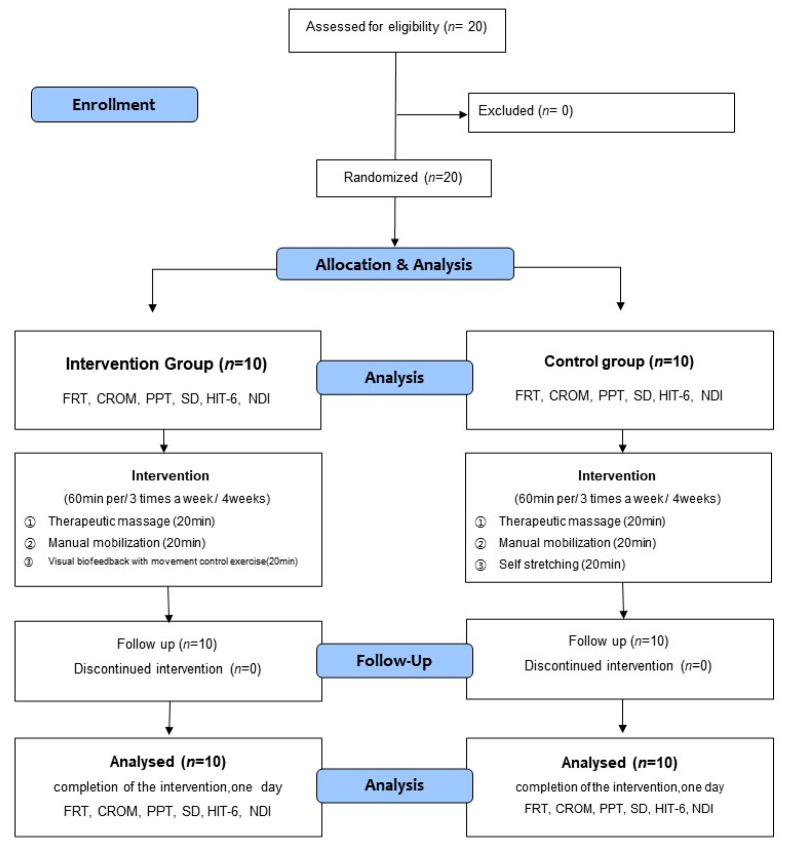
Flow diagram of study.

**Figure 2 healthcare-11-01439-f002:**
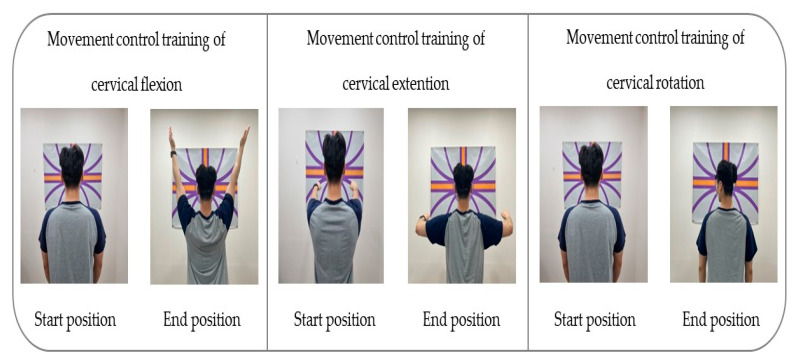
Movement control exercise using a laser device.

**Figure 3 healthcare-11-01439-f003:**
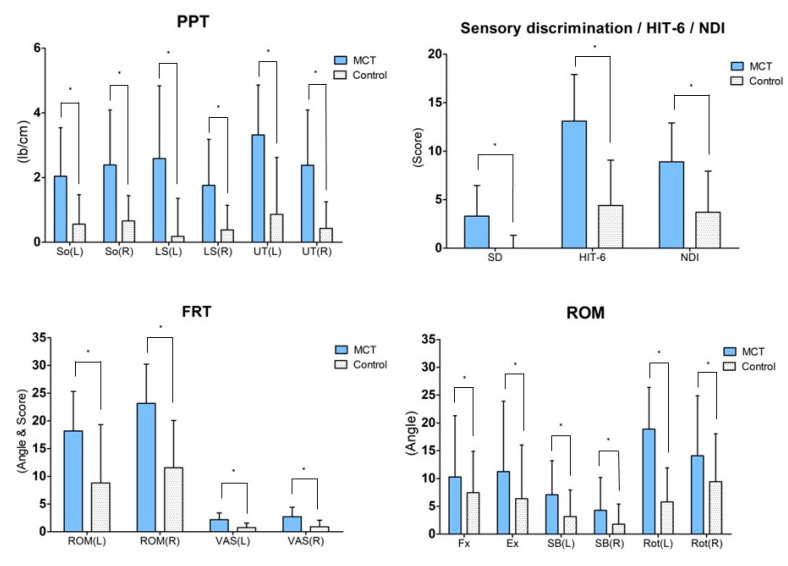
Result of post-hoc analysis between both groups. Values are means ± SD. * Significant difference between groups (*p* < 0.05). Abbreviations: MCT: Movement control training using a laser device; So: Subocciptalis; LS: Levator scapulae; UT: Upper trapezius; HIT-6: Headache impact test-6; NDI: Neck disability index; FRT: Flexion rotation test; ROM: Range of motion; VAS: Visual analog scale; Fx: Flexion; Ex: Extension; SB: Side bending; Ro: Rotation.

**Table 1 healthcare-11-01439-t001:** The intervention program by group.

MCT Group	Control Group
1. Therapeutic massage (20 min)	1. Therapeutic massage (20 min)
① Suboccipiatlis ② Levator scapulae ③ Upper trapezius	① Suboccipiatlis ② Levator scapulae ③ Upper trapezius
2. Manual mobilization (20 min)	2. Manual mobilization (20 min)
① C0-C1 ② C2-C3	① C0-C1 ② C2-C3
3. Movement control exercise using a laser device (20 min)	3. Self stretching (20 min)
① MCT of cervical flexion ② MCT of cervical extension ③ MCT of rotation	① Levator scapulae ② Upper trapezius

MCT: Movement control training using a laser device.

**Table 2 healthcare-11-01439-t002:** Demographic and anthropometric characteristics of the subjects (n = 20).

Variables	MCT (*n* = 10)	Control (*n* = 10)	*t*	*p*
Gender (male/female)	2/8	3/7	-	-
Age (years)	27.60 ± 3.40	25.00 ± 3.49	−1.72	0.085
Height (cm)	164.00 ± 6.32	164.6 ± 6.25	−0.76	0.940
Body mass (kg)	58.00 ± 9.91	59.20 ± 9.35	−0.724	0.469
BMI (%)	21.47 ± 2.58	21.77 ± 2.42	−0.265	0.791

*p* < 0.05, Mean ± Standard Deviation. MCT: Movement control training using a laser device; BMI: Body mass index.

**Table 3 healthcare-11-01439-t003:** Outcome (FRT, ROM) measures for pre- and post-test in MCT and control groups before and after treatment.

		Group	Pre-Test	Post-Test	t	*p*	Post-Pre	t	*p*	ES
			M ± SD	M ± SD			**Mean Difference**			
FRT	ROM(L)	MCT	24.97 ± 7.54	43.13 ± 1.99	−8.00	0.000 *	18.17 ± 7.14	2.33	0.032 *	1.043
		Control	29.74 ± 6.74	38.53 ± 6.16	−2.64	0.27 *	8.79 ± 10.52			
	ROM(R)	MCT	20.27 ± 7.22	43.41 ± 2.57	−10.28	0.000 *	23.15 ± 7.08	3.30	0.004 *	1.480
		Control	25.95 ± 5.37	37.48 ± 5.50	−4.26	0.002 *	11.53 ± 8.55			
	VAS(L)	MCT	3.10 ± 1.37	0.93 ± 1.07	5.66	0.000 *	−2.17 ± 1.21	−3.08	0.006 *	1.382
		Control	3.20 ± 1.39	2.47 ± 0.92	2.74	0.023 *	−0.73 ± 0.84			
	VAS(R)	MCT	4.20 ± 1.68	1.50 ± 1.52	4.97	0.001 *	−2.70 ± 1.71	−2.74	0.013 *	1.232
		Control	3.40 ± 1.07	2.52 ± 0.79	2.31	0.046 *	−0.88 ± 1.20			
ROM	Flexion	MCT	52.37 ± 9.42	62.66 ± 4.72	−2.94	0.016 *	10.28 ± 11.02	0.675	0.509	0.301
		Control	50.03 ± 3.08	57.47 ± 11.11	−3.15	0.012 *	7.44 ± 7.45			
	Extension	MCT	58.23 ± 11.26	69.47 ± 11.35	−2.81	0.02 *	11.25 ± 12.64	0.968	0.346	0.432
		Control	57.23 ± 8.27	63.59 ± 11.16	−2.08	0.067	6.38 ± 9.66			
	Side bending(L)	MCT	37.88 ± 3.30	44.96 ± 6.71	−3.67	0.005 *	7.09 ± 6.11	1.60	0.126	0.717
		Control	40.98 ± 4.47	44.13 ± 3.34	−2.08	0.067	3.16 ± 4.77			
	Side bending(R)	MCT	36.19 ± 3.93	40.47 ± 6.4	−2.28	0.048 *	4.27 ± 5.93	1.13	0.271	0.508
		Control	38.87 ± 3.25	41.65 ± 1.64	−1.55	0.154	1.77 ± 3.63			
	Rotation(L)	MCT	53.01 ± 5.48	71.90 ± 8.60	−7.9	0.000 *	18.88 ± 7.52	4.26	0.000 *	1.909
		Control	60.90 ± 3.63	66.70 ± 4.68	−3.00	0.015 *	5.79 ± 6.12			
	Rotation(R)	MCT	57.53 ± 6.06	71.63 ± 9.39	−4.13	0.003 *	14.10 ± 10.78	1.07	0.299	0.478
		Control	58.97 ± 5.45	68.39 ± 7.65	−3.26	0.007 *	9.43 ± 8.61			

* *p* < 0.05, Mean ± Standard Deviation. FRT: Flexion rotation test; ROM: Range of motion; VAS: Visual analog scale; ES: effect size.

**Table 4 healthcare-11-01439-t004:** Outcome (PPT, sensory discrimination) measures for pre- and post-test in MCT and control groups before and after treatment.

		Group	Pre-Test	Post-Test	t	*p*	Post-Pre	t	*p*	ES
			M ± SD	M ± SD			Mean Difference			
PPT	So(L)	MCT	2.67 ± 1.38	4.69 ± 2.55	−4.22	0.002 *	2.04 ± 1.50	2.66	0.016 *	1.193
		Control	2.04 ± 1.09	2.57 ± 0.90	−1.89	0.91	0.56 ± 0.91			
	So(R)	MCT	2.11 ± 1.10	4.50 ± 2.35	−4.43	0.002 *	2.39 ± 1.70	2.91	0.009 *	1.308
		Control	2.01 ± 0.89	2.65 ± 0.71	−0.08	0.029 *	0.66 ± 0.78			
	LS(L)	MCT	4.12 ± 2.37	6.70 ± 4.01	−3.60	0.006 *	2.59 ± 2.25	2.99	0.008 *	1.341
		Control	3.41 ± 1.53	3.59 ± 1.57	0.66	0.642	0.18 ± 1.18			
	LS(R)	MCT	3.62 ± 2.37	5.38 ± 2.19	−3.91	0.004 *	1.76 ± 1.42	2.70	0.017 *	1.211
		Control	3.59 ± 1.57	3.66 ± 1.56	0.14	0.134	0.38 ± 0.76			
	UT(L)	MCT	3.10 ± 1.89	6.43 ± 2.70	−6.79	0.000 *	3.32 ± 1.54	3.31	0.004 *	1.487
		Control	2.84 ± 1.54	3.70 ± 2.60	0.40	0.157	0.86 ± 1.76			
	UT(R)	MCT	3.00 ± 2.13	5.38 ± 2.19	−4.38	0.002 *	2.38 ± 1.71	3.23	0.005 *	1.454
		Control	3.23 ± 1.44	3.66 ± 1.56	0.16	0.136	0.43 ± 0.82			
Sensory discrimination		MCT	17.60 ± 2.83	20.9 ± 1.52	−3.29	0.009 *	3.30 ± 3.16	3.03	0.007 *	1.361
		Control	17.7 ± 2.00	17.7 ± 1.33	0.000	1.00	0.00 ± 1.33			

* *p* < 0.05, Mean ± Standard Deviation. MCT: Movement control training using a laser device; PPT: pressure pain threshold So: Subocciptalis; LS: Levator scapulae; UT: Upper trapezius; ES: effect size.

**Table 5 healthcare-11-01439-t005:** Outcome(HIT-6, NDI) measures for pre- and post-test in MCT and control groups before and after treatment.

	Group	Pre-Test	Post-Test	t	*p*	Post-Pre	t	*p*	ES
		M ± SD	M ± SD			Mean Difference			
HIT-6	MCT	57.50 ± 2.54	44.40 ± 4.40	8.59	0.000 *	13.10 ± 4.81	−4.10	0.001 *	1.835
	Control	55.80 ± 2.44	51.40 ± 4.90	2.97	0.015 *	4.40 ± 4.67			
NDI	MCT	11.60 ± 3.45	5.30 ± 2.75	5.84	0.000 *	8.90 ± 4.01	−2.81	0.011 *	1.260
	Control	10.70 ± 3.40	7.00 ± 2.66	2.75	0.022 *	3.70 ± 4.24			

* *p* < 0.05, Mean ± Standard Deviation. MCT: Movement control training using a laser device; HIT-6: Headache impact test-6; NDI: Neck disability index; ES: effect size.

## Data Availability

The data presented in this study are available on request from the corresponding author.

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
