# Peer review of "Impact of Movement Control Training Using a Laser Device on the Neck Pain and Movement of Patients with Cervicogenic Headache: A Pilot Study"

_healthcare, 2023, doi:10.3390/healthcare11101439_

Round 1

Reviewer 1 Report

Recommendation

Minor Revision

Report

In this study, the authors examined the impact of movement control training using a laser device on the neck pain and movement of patients with cervicogenic Headache. This article is well written. It can be accepted as this current version. However, the following minor concerns should be addressed and revised.

1.     The values of VAS( R) MCT (25.95±5.37) in Table 3 are wrong, please check it and correct it.

2.     The paragraph (line 344-349) could be combined into the paragraph (line 329-331). It would make them logical and readable when the authors discussed pain in this article.

Author Response

Thank you for your valuable comments. Based on your suggestions, we have done our best to reflect on and amend the review results.
The content of the revised is attached in the attachment below.

Reviewer 2 Report

Dear authors,

Thank you for your work. CGH is a relevant disorder to look into within the domain of rehabilitation. I read the introduction and methodology of your study. I'm afraid that in the current form, the study is not mature, and innovative to add new information to the field. I therefore advise to reject it. I provide you my comments/reflections. Maybe this can be helpful in the future.

INTRODUCTION

Line 32-33. Please look into the literature. CGH is not the most common type of headache. Prevalence of migraine/tension-type headache is a lot higher. Please revise.

Line 40-43. Please revise the grammar. I advise to rephrase ‘negative stimulation of the sensory receptors’. I’m afraid this is difficult to understand, I actually struggle with the phrasing myself.

Line 41-43. This sentence is difficult to interpret. Please rephrase.

Line 42-43. I would be more specific (Castien and De Hertogh, 2019)on the symptoms of CGH. Mostly convergence occurs at the V1 – ophthalmic part of the trigeminal nerve.

Line 48. Careful. Is it proven that posture causes muscle weakness? Are you sure of a causative relation? The references provided do not support this statement. Please adapt.

Line 47-51. A grammar check is needed.

Line 53. The authors indicate that muscle tension causes CGH. Yet, the authors stated that a cervical disorder is related to CGH. I think this needs some explanation. A muscular problem is still referred to as tension-type headache (ICHD, Bendtsen et al). I advise to clarify this matter. Starting with the definition of CGH.

Line 53-55. Please check this sentence for missing words.

Line 56. What does ‘reduced’ mean? I think this verb has not been used correctly. Please revise.

Line 59. Please revise. The message is not clear. What does ‘basic postural improvement’ mean?

Line 46-59. This paragraph seems more associated with tension-type headache? Please clarify.

Line 67. The authors refer to movement control training as a new method. I’m afraid I must disagree. This is not new, nor innovative. The supportive reference is outdated (2012). Please try to convince the reader why such movement control training would be of value in CGH.

Line 69. Please define: ‘functional component’, ‘physical, psychological, behavioural aspect’.

General. The objective of the study needs to be supported better. Stronger arguments, recent literature is needed. At this point the Introduction does not provide the necessary background. Why would one use movement control training based on a study from 2012, in patients with chronic neck pain ..

Why should a habitual pattern being corrected? And, what is a habitual pattern?

Recent literature on spinal pain is leaving the path of corrective therapy. What is the reason for the authors to use this intervention? Research is evolving towards ‘change in posture’ rather than ‘corrective exercises’. Please reflect on this.

MATERIALS AND METHODS

The participants were treated 3 times a week. This seems a lot from a clinical point of view. Given that CGH relates to a cervical disorder, I don’t understand why such high treatment frequency is needed? Can the authors please explain?

Did the authors randomize the groups? And, if yes, how was this applied?

Line 115-116. Please define ‘neck ROM below normal range’. What is normal? Since the authors did not use age or gender as eligibility criteria, it is difficult to interpret normal?

Line 119-123. I’m afraid the eligibility criteria are causing a major problem. What about age, gender, BMI, trauma, smoking, central pain facilitation, … As for now, your sample is not well defined. Please reflect on this.

Line 123. How did the authors end with a sample of 20? Which primary outcome (and reference) were used to determine the sample size? Did you look at means (SD), comparing 2 groups by t-tests? It seems the study is majorly underpowered.

Line 128-… . Who performed the test? Only one rater?

Please provide psychometric references (e.g. reliable, valid, ICCs, …) to support the measurements.

Was the FRT only measured pre-test or also post-test? Please clarify.

Line 149- .. Which algometer was used (brand, type)? The algometer should be perpendicularly applied, not just ‘vertical’. Which testing chronology was applied? Did the authors perform a test before actually measuring?

Also, provide psychometric data (also for the NDI and HIT-6).

Major reflections.

1.  The objective is to correct movement control. Yet, at this point I don’t know if this was even needed? Is movement control decreased in your sample? Did you measure this? If not, why training control if you don’t know if is actually an issue? Wouldn’t this be a ‘one-size-fits all approach’. What is your hypothesis?

2. The authors are measuring a lot of variables in a really small sample (n=10/group). I’m afraid results will be overestimated.

Line 179- … . Intervention. I worry that if everyone received an identical intervention, without determining the individual need, you might create a major issue. Mobilisations are applied both at the L and R side? Why? Why is everyone receiving a massage? wouldn’t all these interventions facilitate pain mechanisms? How much time did the authors provide between interventions? As for now, it is stated ‘an adequate amount of resting time’ this is really too vague.  

The intervention (movement control) is based on literature from 2012. Please revise. The authors also use the neutral posture. A major debate is ongoing concerning such posture. How to determine, and is it even relevant? I think some discussion is needed between the authors on how to fix this.

Author Response

(The authors gave the same response as above.)

Reviewer 3 Report

The work is interesting, congratulations on the idea.  However, I have the following comments.

1.     L79 - Most previous studies - Add citations at the end of the sentence. The authors refer to multiple studies without citing any.

2.     L84 - Add a research hypothesis.

3.     L150 - This is too much of a generality one should describe the exact location of the surveyed points.

4.     Materials and Methods - In this section, I did not find clear information about when the patients were examined after the therapy, what period of time passed. I suggest this information be added to the figure1 or  table1.

5.     In my opinion, statistics are basic. The effect size should be added.

doi: 10.4300/JGME-D-12-00156.1   DOI: 10.11613/BM.2016.015

I would also suggest adding a confidence interval.

6.     L233 - Suggests not to repeat ''(p<.05)'' suggests adding exact values.

7.     L251 –‘’ >’’  - The marker is a different color. Isn't this an error?

8.     L282 - After adding the hypothesis, determine whether it was confirmed. Briefly cite the most important results.

9.     L373-382 - I would suggest adding connections between the neck muscles and the head muscles. Works relevant to the article:

DOI: 10.3390/diagnostics11040580

DOI: 10.3390/ijerph19031577

I propose to add them to the discussion and in their context develop connections between the cervical segment and head muscles that can be related to the results of the your paper.

10.  L398 - Suggests rewriting the beginning of the conclusions, they are too much a repetition of the results.

11.  Due to the significant number of abbreviations used, he suggests adding a list of all abbreviations at the end of the article.

12.  L410 – ‘’ ization, S.B..’’ is in a different font.

13.  References

a.      #2 – ‘’ , HCCotIHSJ’’ - Error in quote.

b.     #6, #15, #29, #45, #49, #56- incomplete quotes.

c.      60% of articles are older than 10 years, suggests adding newer publishes.

14.  Main comment

In the questionnaire used by the authors of the Headache Impact Test-6, there are questions about changes over ''4 weeks.'' Please indicate after what time the authors retested the patients, according to my earlier comment. Regarding the required 4 weeks period imposed by the questionnaire, I have the following question. L111 - Pain and strain in the cervical spine may have a primary cause in a refractive defect (such as myopia) or poor correction of that defect.

Please answer

1.     did the patients have a refractive defect?

2.     if so, did they change lens/glasses powers during the 4 weeks ?

According to recent studies, a refractive defect affects muscles - including neck muscles. Changes in correction can affect this. 

·       PMID: 16646640

·       DOI: 10.3390/ijerph20054524

No changes in muscle activity were observed in subjects without refractive defects.

·       PMID: 25002919

In summary, if patients had a refractive defect and for 4 weeks there was a correction of this lesion. There may have been tonification of cervical tension.

Author Response

(The authors gave the same response as above.)

Round 2

Reviewer 3 Report

The work after the authors' corrections in my opinion is acceptable. Thank you for the opportunity to review again.
With best regards

Author Response

(The authors gave the same response as above.)
